# EntProp: High Entropy Propagation via Auxiliary Batch Normalization Layers

## Abstract

Deep neural networks (DNNs) struggle to generalize to out-of-distribution domains that are different from those in training despite their impressive performance. In practical applications, it is important for DNNs to have both high standard accuracy and robustness against out-of-distribution domains. One technique that achieves both of these improvements is disentangled learning with mixture distribution via auxiliary batch normalization layers (ABNs). This technique treats clean and transformed samples as different domains, allowing a DNN to learn better features from mixed domains. However, if we distinguish the domains of the samples based on entropy, we find that some transformed samples are drawn from the same domain as clean samples, and these samples are not completely different domains. To generate samples drawn from a completely different domain than clean samples, we hypothesize that transforming clean high-entropy samples to further increase the entropy generates out-of-distribution samples that are much further away from the in-distribution domain. On the basis of the hypothesis, we propose high entropy propagation (EntProp), which feeds high-entropy samples to the network that uses ABNs. We introduce two techniques, data augmentation and free adversarial training, that increase entropy and bring the sample further away from the in-distribution domain. These techniques do not require additional training costs. Our experimental results show that EntProp achieves higher standard accuracy and robustness with a lower training cost than existing methods. In particular, EntProp is highly effective at training on small datasets.

## 1 Introduction

Deep neural networks (DNNs) have achieved impressive performance in a variety of fields, such as computer vision, natural language processing, and speech recognition. However, DNNs are susceptible to accuracy degradation when presented with data distributions that deviate from the training distribution. This is a common occurrence in outdoor environments, such as autonomous driving and surveillance cameras, due to variations in weather and brightness (Diamond et al., 2021; Hendrycks & Dietterich, 2019; Zendel et al., 2018). As a result, while standard accuracy is essential for DNNs, robustness against distribution shifts is equally important.

Various techniques have been proposed to improve the robustness against out-of-distribution domains (*e.g.*, domain adaptation (Saenko et al., 2010; Ganin & Lempitsky, 2015; Tzeng et al., 2015)), many of which usually decrease the standard accuracy. One technique to improve both standard accuracy and robustness is disentangled learning with mixture distribution using a dual batch normalization (BN) layer (Xie et al., 2020; Mei et al., 2022; Zhang et al.; Wang et al., 2021). This technique prepares an auxiliary BN layers (ABNs) in addition to the main BN layers (MBNs). It feeds the clean samples and the samples transformed by adversarial attacks or data augmentation to the same network but applied with different BNs, *i.e.*, use the MBNs for the clean samples and use the ABNs for the transformed samples. The distinction of the BNs used to train samples of different domains prevents mixing of the BN layer statistics and the affine parameters (Zhang et al., 2023), allowing the MBN-applied network to learn better from the features of both the out-of-distribution and in-distribution domains (Xie et al., 2020). Furthermore, since only MBNs are used during inference, there is no increase in computational cost in test-time.

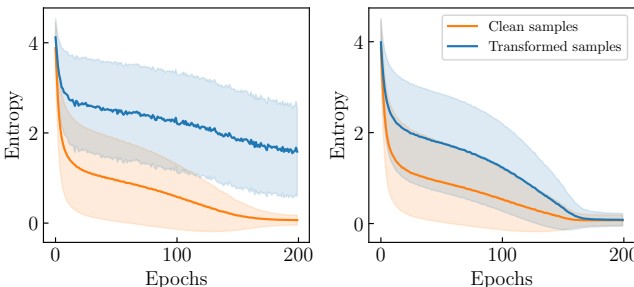

Figure 1: Entropy per epoch when ResNet-18 is trained with MixProp (Zhang et al.) (left) and AdvProp (Xie et al., 2020) (right) on the CIFAR-100 dataset. Error bars indicate one standard deviation, and lines indicate average.

Existing studies treat clean and transformed samples as different domains; however, it is not clear whether these samples are entirely different domains. It is clear that clean samples are in-distribution domain. The transformed samples can be divided into two groups: those that are highly transformed and those that are less transformed. Therefore, we have the following research questions: *Do transformed samples include samples drawn from both the in-distribution and out-of-distribution domains?*

As a first step in answering this question, we consider distinguishing between the in-distribution and out-of-distribution samples. Since adversarial attacks and data augmentation are transformations that increase the diversity and hardness of samples (Wang et al., 2021), we verify the distinction of domains using an uncertainty metric, entropy. Figure 1 shows the entropy of clean and transformed samples when training the network with existing methods. The results show that some of the clean samples with high entropy overlap with the entropy of the transformed samples. Since clean high-entropy samples are already similar to out-of-distribution samples, we hypothesize that applying entropy-increasing transformations to clean high-entropy samples generates out-of-distribution samples that are much further away from the in-distribution samples. From this hypothesis, we propose high entropy propagation (EntProp), which trains ABN-applied network with high-entropy samples. First, a network trains clean samples using MBNs and calculates entropy. Then, for the high-entropy samples in the clean samples, a network trains using ABNs. At this time, to further increase the entropy of the samples and bring them further away from the in-distribution domain, we introduce two techniques, data augmentation and free adversarial training (Shafahi et al., 2019). These techniques have no additional training cost and allow for further accuracy gains.

We evaluated EntProp on two image classification datasets, CIFAR-100 (Krizhevsky et al., 2009) and CUB-200-2011 (Welinder et al., 2010), with several DNN architectures. We show that training ABN-applied networks on clean high-entropy samples improves both standard accuracy and robustness even though it does not use adversarial attacks or data augmentation. EntProp, which includes two entropy-increasing techniques, shows higher accuracy at a lower training cost than comparison methods. Furthermore, we show that on the small dataset, the use of adversarial training on all samples leads to overfitting and accuracy degradation, which can be resolved by undersampling, such as EntProp.

The contributions of this paper are as follows:

- We propose EntProp, a novel disentangled learning method via ABNs. We treat high-entropy samples as out-of-distribution domain, and introduce two techniques to further separate them from the in-distribution domain.

- Our experiments show that EntProp achieves better standard and robustness than existing methods, despite its lower training cost.

- We demonstrate that using all samples for adversarial training on small datasets leads to overfitting and lower accuracy than vanilla training. Undersampling methods such as EntProp prevent overfitting, benefit from adversarial training, and improve accuracy.

## 2    RELATED WORK

Adversarial attacks (Goodfellow et al., 2015; Madry et al., 2018) cause DNNs to make wrong predictions by adding human imperceptible perturbations to input sample. To defend against such attacks, a variety of methods (Kannan et al., 2018; Zhang et al., 2019; Wang et al., 2020) have been proposed to train DNNs with adversarial samples, also known as adversarial training. However, adversarial training has a trade-off (Tsipras et al., 2018; Ilyas et al., 2019) between accuracy on clean samples and robustness to adversarial attacks, compromising accuracy on clean samples in order to achieve high robustness. The reason for this trade-off was thought to be that the two domains are learned simultaneously by a single DNN, motivated by the two-domain hypothesis (Xie & Yuille, 2019) that clean and adversarial samples are drawn from different domains. Based on this hypothesis, Xie & Yuille (2019) showed that using MBNs for clean samples and ABNs for adversarial samples avoids mixing the statistics and affine parameters of BN layers (Zhang et al., 2023) by two different domains and achieves high accuracy for the domain for which each BN layer is trained. AdvProp (Xie et al., 2020) showed that disentangled learning for a mixture of distributions via ABNs allows DNNs with MBNs to learn more effectively from both adversarial and clean samples, improving the standard accuracy and the accuracy for the out-of-distribution domain. AdvProp is simple and highly practical, and has since been developed in various ways. Fast AdvProp (Mei et al., 2022) reduced the number of samples and iterations required for adversarial attacks, resulting in the same computational cost as vanilla training, with higher accuracy. Disentangled learning via ABNs showed effectiveness not only using adversarial attacked samples, but also using data augmented samples (Merchant et al., 2020; Zhang et al.; Wang et al., 2021) and style transferred samples (Li et al., 2020). Furthermore, AdvProp was proposed for various applications, including object detection tasks (Chen et al., 2021), contrastive learning (Jiang et al., 2020; Ho & Nvasconcelos, 2020), and training vision transformers (Herrmann et al., 2022).

Although these studies treat clean and transformed samples as different domains, we argue that some of these samples overlap in domain. We train the MBN-applied network with clean samples as the in-distribution domain and the ABN-applied network with high-entropy samples as the out-distribution domain.

## 3    PROPOSED METHOD

In this section, we describe our method, high entropy propagation (EntProp), for effective disentangled learning with mixture distribution via ABNs.

### 3.1    MOTIVATION

Existing methods treat clean samples as the in-distribution domain and samples transformed by adversarial attacks (Xie et al., 2020; Mei et al., 2022; Xie & Yuille, 2019) or data augmentation (Zhang et al.; Merchant et al., 2020) as the out-of-distribution domain, and distinguish the BNs used for these samples. Although it is clear that clean samples are the in-distribution domain, we question that transformed samples are the out-distribution domain. In the transformed samples, some samples are significantly affected by the transformation and are further away from the in-distribution domain, while some samples are less affected and are closer to the in-distribution domain. If the distribution is distinguished by entropy as shown in Figure 1, some samples in the clean and transformed samples have overlapping domain, which may prevent effective disentangled learning via ABNs. Since clean high-entropy samples are in the same domain as the transformed out-of-distribution samples, we hypothesize that transforming these samples to increase entropy generates out-of-distribution samples that are significantly different from the in-distribution samples. On the basis of the hypothesis, we propose EntProp, which trains the ABN-applied network on high-entropy samples.

### 3.2    METHODOLOGY

Here, we describe the process of one iteration of EntProp training. We assume a network with ABNs in addition to the MBNs. Figure 2 shows the overview of EntProp and existing methods, and Algorithm 1 shows the pseudo-code of EntProp.

Table 1: Training costs for each method. $p_{\text{adv}}$ is the Fast AdvProp hyperparameter that determines the sample percentage used for adversarial attack.

|  | Vanilla | AdvProp | Fast AdvProp | MixProp | EntProp |
|---|---|---|---|---|---|
| Training Cost | N | $(2 + n)$N | $(1 + p_{\text{adv}})$N | 2N | $(1 + kn)$N |

**Sample Selection.** First, the MBN-applied network outputs predictions from clean samples. From the predictions, we compute loss and entropy. Next, we feed the top $k|\mathcal{B}|$ samples of high-entropy samples to the ABN-applied network to compute the loss, where $k \in [0, 1]$ is a hyperparameter and $|\mathcal{B}|$ is the batch size. Finally, we update the network parameters from the gradient to minimize total loss. Furthermore, based on our hypothesis, we introduce two techniques that increase entropy of samples without additional training cost: data augmentation and free adversarial training.

**Data Augmentation.** Data augmentation is the most common technique widely used when training DNNs that improves the accuracy of the DNN by transforming samples and increasing diversity and hardness. Since most data augmentations use simple transformations, the computational cost is negligible compared to training DNNs. We use MixUp (Pang et al., 2019), the most typical data augmentation. MixUp linearly combines two samples in a mini-batch and increase entropy because the combined sample has two labels. Unlike MixProp (Zhang et al.), we treat augmented samples as in-distribution domain and train MBN-applied network from the augmented samples for the calculation of loss and entropy. Since MixUp is a method for improving standard accuracy, samples transformed by MixUp retain sufficient information about the in-distribution domain. Furthermore, MixUp eliminates the high-entropy sample selection bias in each iteration, allowing the ABN-applied network to train a diversity of samples (see Appendix for details). The MixUp loss function is defined as:

$$L^m = \lambda L^c(\theta, \boldsymbol{x}^m, y^a) + (1 - \lambda)L^c(\theta, \boldsymbol{x}^m, y^b), \tag{1}$$

where $L^c$ is the cross-entropy loss, $\theta$ is the network parameter, $\lambda$ is the mixing coefficient, $\boldsymbol{x}^m$ is the mixed sample, and $y^a$ and $y^b$ are the labels of the sample before mixing. If EntProp does not use MixUp, the MBN-applied network trains $L^c$ for clean samples.

**Free Adversarial Training.** Shafahi et al. (2019) generates adversarial examples by reusing the gradients used for training in the previous iteration. We use this technique to generate adversarial examples $\boldsymbol{x}^a$ for high-entropy samples. EntProp first calculates the loss to clean or augmented samples with MBN-applied network, allowing the generation of free adversarial examples from the gradient at this time. Note that it is not optimal to use the MBN-applied network gradient to generate an adversarial attack on the ABN-applied network. When we use augmented samples, we use the gradient obtained from the augmentation loss to generate an adversarial example. In the case of multiple iterations for the attacker, as in a Projected Gradient Descent (PGD) (Madry et al., 2018) attack, the first one has no computational cost, but the subsequent ones have the same computational cost as a standard adversarial attack and are generated from the gradient of the ABN-applied network. For the PGD attack, we set perturbation size $\epsilon$ to $n + 1$ and attack step size $\alpha$ to 1, where $n$ is the number of iterations for the attacker. If the number of iterations is 1, then $\epsilon$ is set to 1.

### 3.3 TRAINING COST

Here, we consider the training cost of one epoch. We denote the cost of a single forward and backward pass for a single sample as 1 and the size of the dataset as $N$. The cost of vanilla training for one epoch is $N$. EntProp first uses the clean mini-batch, then $k|\mathcal{B}|$ samples of the mini-batch, thus the cost is $(1 + k)N$. The computational cost of data augmentation and free adversarial training ($n = 1$) is negligible compared to the computational cost of forward and backward passes, thus using them does not change the overall training cost. If we increase the iteration number $n$ of the adversarial attack by more than 1, it cost us an additional $k(n-1)N$. Consequently, the training cost of EntProp is $(1 + kn)N$. Table 1 shows the training cost per epoch for existing methods and EntProp.

### 4 EXPERIMENTS

In this section, we experimented on two image classification datasets: CIFAR-100 (Krizhevsky et al., 2009) and CUB-200-2011 (Welinder et al., 2010), and show the effectiveness of EntProp.

---

**Algorithm 1:** Pseudo code of EntProp

---

**Data:** A set of clean samples with labels;
**Result:** Network parameter $\theta$;
**for** *each training step* **do**
    Sample a clean mini-batch $\boldsymbol{x}$ with label $y$;
    Generate the corresponding augmented mini-batch $\boldsymbol{x}^m$ and labels $y^a$ and $y^b$;
    Compute loss $L^m$ and entropy on augmented mini-batch using the MBNs;
    Obtain the gradient $\nabla \leftarrow \nabla_{\boldsymbol{x}^m}$;
    Get the top$k|\mathcal{B}|$ samples $\boldsymbol{x}^a$ with the highest entropy from augmented mini-batch;
    $\delta \leftarrow 0$;
    **for** $i = 1, \ldots, n$ **do**
        $\delta \leftarrow \delta + \epsilon \cdot sign(\nabla)$;
        $\boldsymbol{x}^a = \boldsymbol{x}^a + clip(\delta, -\epsilon, \epsilon)$;
        Compute loss $L^c(\theta, \boldsymbol{x}^a, y)$ on adversarial sample using the ABNs;
        Obtain the gradient $\nabla \leftarrow \nabla_{\boldsymbol{x}^a}$;
    **end**
    Minimize the total loss w.r.t. network parameter $\arg\min_{\theta} L^m + L^c(\theta, \boldsymbol{x}^a, y)$.
**end**
**return** $\theta$

---

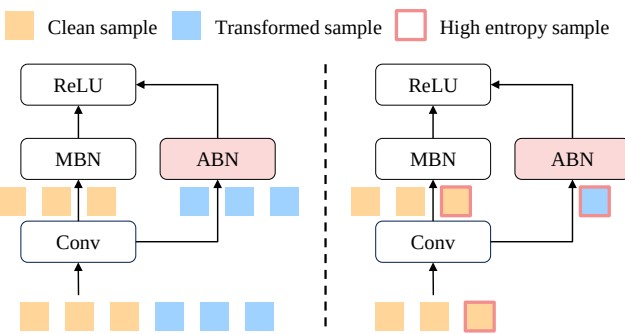

Figure 2: Overview of existing methods (left) and EntProp (right). The existing methods feed clean samples to MBN and transformed samples to ABN. EntProp feeds clean samples to MBN and transforms clean high-entropy samples and feeds them to ABN.

## 4.1 EXPERIMENTS SETUP

### 4.1.1 DATASETS AND ARCHITECTURES

**CIFAR-100.** CIFAR-100 dataset consists of 50000 training images and 10000 test images, with 100 classes. We used ResNet (He et al., 2016), Wide ResNet (Zagoruyko & Komodakis, 2016), and ResNeXt (Xie et al., 2017) as DNN architectures, and SGD with momentum 0.9 as optimizer. We trained DNNs with batch size of 128 for 200 epochs with weight decay of 0.0005. The learning rate started with 0.1 and decreased by cosine scheduler.

**CUB-200-2011.** CUB-200-2011 dataset consists of 5994 training images and 5794 test images, with 200 classes. We used the ResNet (He et al., 2016) and the EfficientNet (Tan & Le, 2019) family pretrained (maintainers & contributors, 2016) by ImageNet dataset (Russakovsky et al., 2015) as DNN architectures, and Adam (Kingma & Ba, 2015) as optimizer. Since the pretrained networks do not have ABNs, we set the initial weights of the ABNs to be the same as those of the MBNs. We fine-tuned networks with batch size of 64 for 100 epochs with weight decay of 0.0005. The learning rate started with 0.0001 and decreased by the factor of 0.1 at every 10 epochs.

### 4.1.2 EVALUATION.

To evaluate the balance between standard accuracy (SA), which is the accuracy of a standard test set, and robust accuracy (RA), which is the average accuracy of an artificially corrupted test set (Hendrycks & Dietterich, 2019), we define the harmonic mean as our evaluation metric.

$$H_{score} = \frac{2SA \cdot RA}{SA + RA}. \tag{2}$$

The corrupted test set consists of 15 types of corruption[1] with five severity levels, and we use the average accuracy of all of them as RA. $H_{score}$ is high only when both SA and RA are high. All experiments were performed three times and we report the average values.

### 4.1.3 COMPARISON METHODS.

We compared the four methods with EntProp.

- **Vanilla.** Vanilla training for network without ABNs.
- **AdvProp.** AdvProp feeds the clean samples and the adversarial samples to the same network but applied with different BNs. We used PGD as the attacker to generate adversarial samples. We set the perturbation size $\epsilon$ to 4. The number of iterations for the attacker is $n = 5$ and the attack step size is $\alpha = 1$.
- **Fast AdvProp.** Fast AdvProp speeds up AdvProp by reducing the number of iterations for PGD attacker and the percentage of training samples used as adversarial examples. We set the percentages of training samples used as adversarial examples to $p_{adv} = 0.2$, the perturbation size $\epsilon$ to 1, the number of iterations for the attacker to $n = 1$, and the attack step size to $\alpha = 1$.
- **MixProp.** MixProp feed the clean samples and the augmented samples with MixUp to the same network but applied with different BNs. The parameter of the beta distribution used for MixUp is set to 1.

### 4.2 MAIN EXPERIMENTS

In this section, we show the effectiveness of EntProp. For fair comparison, all methods used MixUp during training. All methods except MixProp trained the MBN-applied network on the augmented samples. Fast AdvProp and AdvProp generated adversarial examples from clean samples following the Fast AdvProp setting. We describe more detailed experimental results in the Appendix.

### 4.2.1 EXPERIMENTS ON THE CIFAR-100 DATASET

First, we confirmed the effect of feeding clean high-entropy samples to the ABN-applied network. Figure 3 shows the results of sample selection with high entropy versus random selection. There is little difference when $k$ is small and large, and entropy shows higher $H_{score}$ than random when $k = 0.2$ to $k = 0.7$. Furthermore, $k \geq 0.1$ shows a higher $H_{score}$ than vanilla training ($k = 0$), meaning that the use of ABN is effective. The use of ABN increases the number of network parameters during training and allows the network to achieve good generalization performance.

Next, we verified each component of EntProp to confirm the effect of increasing entropy. We set $k = 0.2$ and $n = 1$. Table 2 shows the results. Training clean high-entropy samples with the ABN-applied network improves both SA and RA from vanilla training even though no additional processing, such as adversarial attacks, is performed. MixUp further improves both SA and RA, while free adversarial training further improves RA but slightly decreases SA. EntProp which uses all components achieves the highest $H_{score}$. Increasing entropy brings the sample further away from the in-distribution domain, allowing effective disentangled learning with mixture distribution. Moreover, Figure 4 shows the entropy of the clean and transformed samples when training the network with EntProp. The results show that EntProp ($k = 0.2, n = 5$) completely distinguishes between the domains of clean and transformed samples, as we hypothesize.

---

[1]Gaussian noise, Shot noise, Impulse noise, Defocus blur, Glass blur, Motion blur, Zoom blur, Snow, Frost, Fog, Brightness, Contrast, Elastic transform, Pixelate, JPEG.

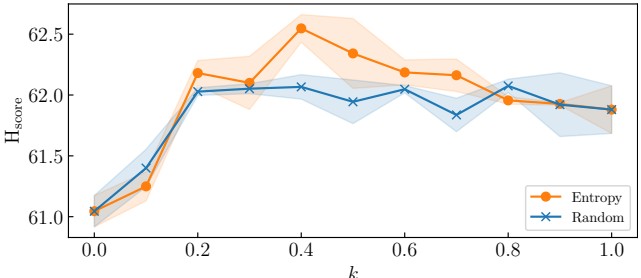

Figure 3: Comparison of high-entropy sample selection to random selection using ResNet-18 on the CIFAR-100 dataset. Error bars indicate one standard error, and lines indicate the average. $k = 0$ is the same as vanilla training, and $k = 1$ feeds all samples to the ABN-applied network.

Table 2: Ablation study with ResNet-18 on the CIFAR-100 dataset. The best and second results are **bolded** and underlined. The numbers in parentheses indicate the differences from vanilla training.

| Sample Selection | MixUp | Free ($n = 1$) | SA(%) | RA(%) | $H_{score}$ |
|:---:|:---:|:---:|:---:|:---:|:---:|
| ✓ | | | 79.24(0.79) | 51.17(1.21) | 62.18(1.14) |
| ✓ | ✓ | | **79.66**(1.21) | 54.53(4.57) | 64.74(3.70) |
| ✓ | | ✓ | 78.55(0.10) | 52.99(3.03) | 63.29(2.25) |
| ✓ | ✓ | ✓ | 79.41(0.96) | **55.24**(5.28) | **65.15**(4.11) |

Then, we compared our method to comparison methods using four network architectures on the CIFAR-100 dataset. Table 3 shows the $H_{score}$ and training cost of the network when trained from scratch by each method. The comparison methods consistently improve $H_{score}$ by adding MixUp to the training. The results show that bringing the samples further away from the in-distribution domain by increasing entropy is highly effective for disentangled learning. EntProp ($k = 0.2, n = 1$) has the same training cost as Fast AdvProp and higher average $H_{score}$. EntProp ($k = 0.4, n = 1$) has a significantly lower training cost than AdvProp and higher average $H_{score}$, and EntProp ($k = 0.6, n = 5$) has a lower training cost than AdvProp + MixUp and the highest average $H_{score}$. These results indicate that EntProp allows for more efficient training by bringing the samples fed to the ABN-applied network further away from the in-distribution domain.

### 4.2.2 EXPERIMENTS ON THE CUB-200-2011 DATASET

We fine-tuned the pre-trained models by each method on the CUB-200-2011 dataset. Table 4 show the results for the ResNet and EfficientNet families, respectively. EntProp ($k = 0.2, n = 5$) has the best results for ResNet-18 and AdvProp for the rest of the ResNet family. The average results of ResNet family show that EntProp ($k = 0.2, n = 1$) outperforms comparison methods with similar training costs, and EntProp ($k = 0.2, n = 5$) has the best results. The experiments with the

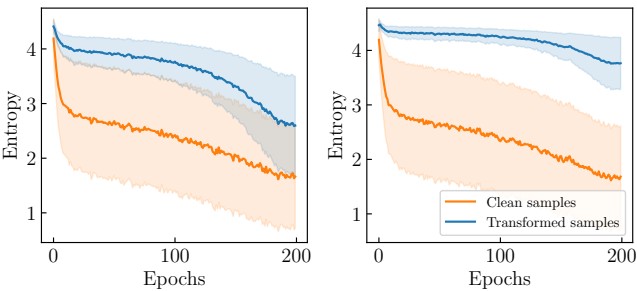

Figure 4: Entropy per epoch when ResNet-18 is trained with EntProp ($k = 0.2, n = 1$) (left) and EntProp ($k = 0.2, n = 5$) (right) on the CIFAR-100 dataset. Error bars indicate one standard deviation, and lines indicate average.

Table 3: $H_{score}$ and training cost for the four network architectures on the CIFAR-100 dataset. Avg. indicates the average of the four networks.

| Method | Cost | ResNet-18 | ResNet-50 | WRN-50 | ResNeXt-50 | Avg. |
|---|---|---|---|---|---|---|
| Vanilla | N | 61.04 | 62.08 | 62.56 | 63.99 | 62.42 |
| Fast AdvProp | 1.2N | 63.63 | 64.45 | 65.25 | 65.17 | 64.63 |
| AdvProp | 7N | 64.69 | 67.17 | 67.17 | 67.37 | 66.60 |
| Vanilla + MixUp | N | 64.11 | 64.19 | 66.61 | 67.93 | 65.71 |
| MixProp | 2N | 64.73 | 66.18 | 66.87 | 67.36 | 66.29 |
| Fast AdvProp + MixUp | 1.2N | 64.59 | 66.83 | 66.74 | 67.39 | 66.39 |
| AdvProp + MixUp | 7N | 65.08 | **69.21** | **69.60** | 69.90 | 68.45 |
| EntProp ($k = 0.2, n = 1$) | 1.2N | 65.15 | 65.93 | 67.00 | 68.08 | 66.54 |
| EntProp ($k = 0.4, n = 1$) | 1.4N | 65.37 | 66.11 | 68.14 | 67.44 | 66.77 |
| EntProp ($k = 0.6, n = 1$) | 1.6N | 65.40 | 66.95 | 68.34 | 68.72 | 67.36 |
| EntProp ($k = 0.2, n = 5$) | 2N | 64.58 | 66.12 | 67.64 | 68.31 | 66.66 |
| EntProp ($k = 0.4, n = 5$) | 3N | **66.46** | 67.95 | 69.58 | 69.03 | 68.26 |
| EntProp ($k = 0.6, n = 5$) | 4N | 66.30 | 69.12 | 69.52 | **69.96** | **68.73** |

Table 4: $H_{score}$ for the ResNet family (top table) and the EfficientNet family (bottom table) on the CUB-200-2011 dataset.

| Method | ResNet-18 | ResNet-50 | ResNet-101 | ResNet-152 | Avg. |
|---|---|---|---|---|---|
| Vanilla + MixUp | 62.11 | 68.66 | 71.54 | 72.36 | 68.67 |
| MixProp | 62.24 | 67.70 | 70.14 | 71.37 | 67.86 |
| Fast AdvProp + MixUp | 63.26 | 69.75 | 72.81 | 72.95 | 69.69 |
| AdvProp + MixUp | 62.69 | **70.66** | **73.12** | **73.77** | 70.06 |
| EntProp ($k = 0.2, n = 1$) | 63.58 | 69.74 | 72.67 | 73.46 | 69.86 |
| EntProp ($k = 0.2, n = 5$) | **64.21** | 70.17 | 72.59 | 73.59 | **70.14** |

| Method | B0 | B1 | B2 | B3 | B4 | B5 | B6 | B7 | Avg. |
|---|---|---|---|---|---|---|---|---|---|
| Vanilla + MixUp | 68.44 | **72.14** | 72.08 | 72.05 | 73.18 | 74.18 | 75.12 | 74.64 | 72.75 |
| MixProp | 67.89 | 71.27 | 71.74 | **73.32** | **74.4** | 74.28 | 75.23 | 75.06 | 72.92 |
| Fast AdvProp + MixUp | 68.85 | 71.85 | 72.26 | 72.56 | 72.75 | 73.97 | 74.96 | 75.01 | 72.79 |
| AdvProp + MixUp | 66.63 | 68.10 | 68.4 | 68.77 | 70.36 | 70.55 | 71.54 | 71.67 | 69.51 |
| EntProp ($k = 0.2, n = 1$) | 69.24 | 71.98 | **72.64** | 72.60 | 74.28 | **74.54** | 75.42 | 74.97 | **73.22** |
| EntProp ($k = 0.2, n = 5$) | **69.41** | 72.06 | 72.34 | 72.23 | 73.52 | 74.46 | 75.22 | **76.32** | 73.21 |

EfficientNet family show that EntProp ($k = 0.2, n = 1$) has the best average results. PGD attack does not show consistent improvement as in other experiments. AdvProp consistently shows significantly lower results than vanilla training. Although adversarial training requires a large amount of dataset (Schmidt et al., 2018), the CUB-200-2011 dataset is small, and training using PGD attack leads to overfitting (Rice et al., 2020). EntProp and Fast AdvProp prevent overfitting by undersampling, while EntProp outperforms Fast AdvProp due to efficient sampling based on entropy.

### 4.2.3 UNCERTAINTY METRIC

We use entropy as a metric to select the samples that EntProp feeds to the ABN-applied network. We evaluated EntProp ($k = 0.2, n = 1$) when using the following metrics, other than entropy, to distinguish between samples in the in-distribution and out-of-distribution domains.

- **Cross-Entropy** is the distance between the true probability distribution and the predicted probability distribution.

- **Confidence** is the maximum class probability.

- **Logit Margin** is the difference between the maximum non-true class probability and the true class probability.

Table 5: $H_{score}$ for different uncertainty metrics on the CIFAR-100 dataset.

| Metrics | ResNet-18 | ResNet-50 | WRN-50 | ResNeXt-50 | Avg. |
|---|---|---|---|---|---|
| Entropy | 65.15 | 65.93 | 67.00 | 68.08 | 66.54 |
| Cross-Entropy | 64.81 | 64.76 | 67.12 | 68.34 | 66.26 |
| Confidence | **65.48** | **66.47** | **67.36** | 67.23 | **66.63** |
| Logit Margin | 64.84 | 66.18 | 65.71 | **68.53** | 66.31 |

Table 6: Adversarial robustness of ResNet-18 on the CIFAR-100 dataset.

| Sample Selection | MixUp | Free ($n = 1$) | Metric | PGD-20 |
|---|---|---|---|---|
| | | | | 6.14 |
| ✓ | | | Entropy | 6.44 |
| ✓ | ✓ | | Entropy | 4.14 |
| ✓ | | ✓ | Entropy | 10.51 |
| ✓ | ✓ | ✓ | Entropy | 4.71 |
| ✓ | ✓ | ✓ | Cross-Entropy | 4.45 |
| ✓ | ✓ | ✓ | Confidence | 4.59 |
| ✓ | ✓ | ✓ | Logit Margin | 4.42 |
| ✓ | ✓ | ✓ | Random | 5.24 |

Because we use MixUp during training, the true label used by these metrics is the original true label of the sample. Table 5 shows the results. All metrics show no significant differences. The results show that different architectures have different effective metrics.

## 5 LIMITATION

In this paper we focus on improving both standard accuracy and robustness against out-of-distribution domains. We additionally evaluated the robustness against the adversarial attack. We evaluated the accuracy of EntProp variants and vanilla training against PGD-20 attack. Table 6 shows the results. Feeding clean high-entropy samples to the ABN-applied network shows higher adversarial robustness than vanilla training, even though adversarial attacks are not used for training. Free adversarial training significantly improves adversarial robustness, but MixUp significantly decreases it. In the comparison of sample selection metrics, random shows the best results rather than using uncertainty metrics. These results indicate that each component of EntProp designed on entropy is effective in improving standard accuracy and out-of-distribution robustness; however, it is not effective in improving adversarial robustness. If the objective is a different evaluation metric than ours, it is necessary to design an appropriate metric that is different from the entropy.

## 6 CONCLUSION

The existing disentangled learning methods train from mixture distribution by treating clean and transformed samples as different domains, and feeding the former to the MBN-applied network and the latter to the ABN-applied network. However, it is not appropriate to treat the clean and transformed samples as different domains. We found that when we verified the domains of the samples based on entropy, the clean and transformed samples had overlapping regions of domains. We hypothesize that further increasing the entropy of clean high-entropy samples generates samples that are further away from the in-distribution domain. On the basis of the hypothesis, we propose a novel method, EntProp, which feeds high-entropy samples to the ABN-applied network. Our experiments show that EntProp has high accuracy even though its training cost is less than that of existing methods. In particular, experiments on small dataset show that Entprop prevents overfitting against adversarial training and outperforms comparison methods. Our method improves standard accuracy and out-of-distribution robustness, but has limitations with respect to adversarial robustness. This limitation suggests the need to design an optimal domain selection metric for each task.

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

# A   SAMPLE SELECTION BIAS

We verified the bias of the high-entropy sample selection during training. Figure 5 shows the results. At $k = 0.2$, the bias is large and most samples are not selected as high-entropy samples. MixUp eliminates high-entropy sample selection bias.

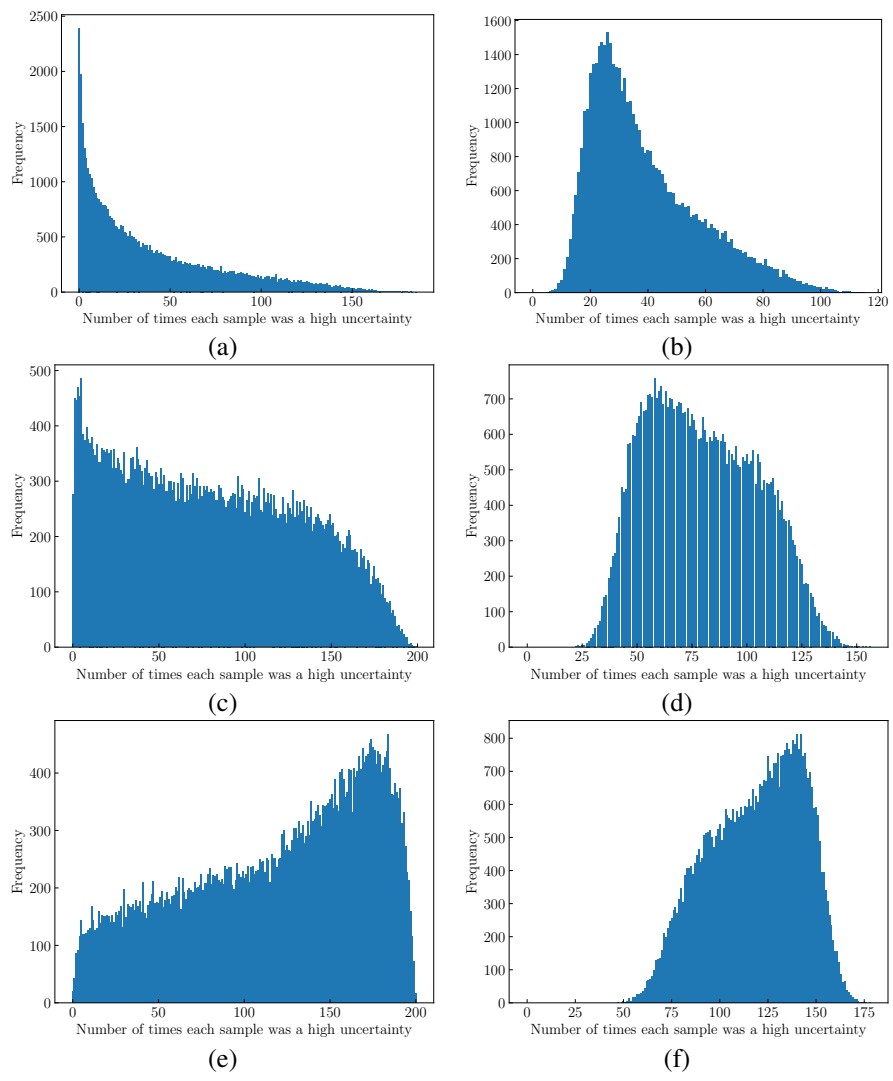

Figure 5:   Histogram of the number of times a sample was selected as a high-entropy sample. (a) $k = 0.2$. (b) $k = 0.2$ w/MixUp. (c) $k = 0.4$. (d) $k = 0.4$ w/MixUp. (e) $k = 0.6$. (f) $k = 0.6$ w/MixUp.

## B  CIFAR-100 EXPERIMENTS DETAIL

Figure 6 shows the entropy of the clean and transformed samples when training the network with EntProp variant. Two techniques show that they increase the entropy of the sample.

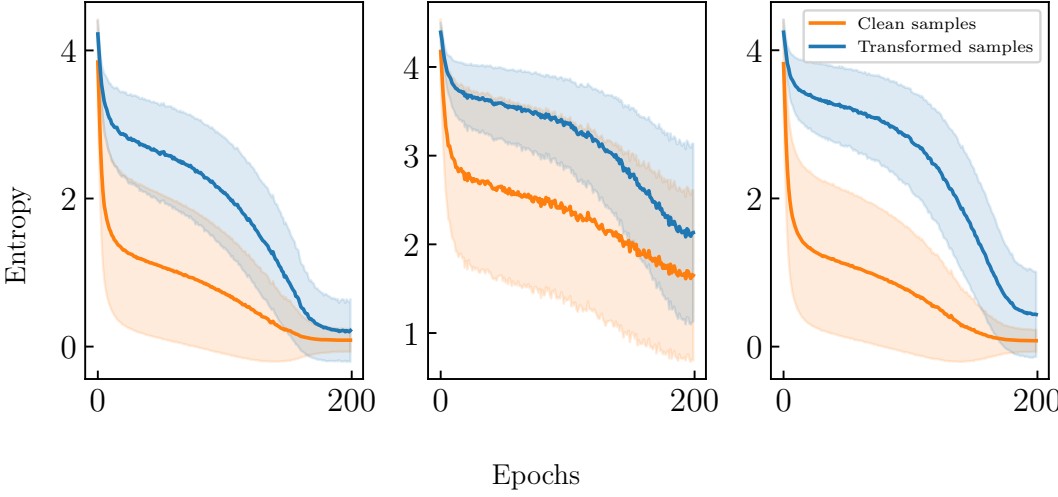

Figure 6:  Entropy per epoch when ResNet-18 is trained with EntProp (w/o MixUp, w/o Free adversarial training) (left), EntProp (w/o Free adversarial training) (center), and EntProp (w/o MixUp) (right) on the CIFAR-100 dataset. Error bars indicate one standard deviation, and lines indicate average.

Table 7 shows the results of standard accuracy and robust accuracy in the CIFAR-100 experiments. MixProp consistently shows the highest SA, but it decreases the RA of WRN-50 and ResNeXt-50. AdvProp shows the highest RA, but decreases SA in many networks. EntProp shows improvement in SA and RA in many cases.

## C  CUB-200-2011 EXPERIMENTS DETAIL

Table 8 shows the results of standard accuracy and robust accuracy for the ResNet family in the CUB-200-2011 experiments. Average results show MixProp with the highest SA and AdvProp with the highest RA. However, MixProp has lower RA and AdvProp has lower SA than vanilla training. EntProp and Fast AdvProp show balanced results that improve both SA and RA.

Table 9 shows the results of standard accuracy and robust accuracy for the EfficientNet family in the CUB-200-2011 experiments. MixProp consistently shows the highest SA, but it decreases the RA in many case. All methods except MixProp show a decrease in SA compared to vanilla training. This result implies that networks trained by adversarial training-based methods are overfitting to adversarial attacks. In particular, AdvProp is more likely to lead to overfitting, with both SA and RA showing lower results than vanilla training. EntProp and Fast AdvProp prevent overfitting and improve RA by reducing the number of samples of adversarial attacks through undersampling. Entropy-based undersampling of EntProp outperforms random sampling of Fast AdvProp.

## D  DATA AUGMENTATION

We compared MixUp and CutMix (Yun et al., 2019) as data augmentations that increase entropy at no additional training cost. Table 10 shows the results. The results show that CutMix outperforms for SA, but MixUp significantly outperforms for RA and $H_{score}$. MixUp, which transforms the entire image, is more likely to increase entropy than CutMix, which transforms a portion of the image, and contributes to improving $H_{score}$.

Table 7: Standard accuracy (top table) and robust accuracy (bottom table) for the four network architectures on the CIFAR-100 dataset. The best and second results are **bolded** and underlined. Avg. indicates the average of the four networks.

| Method | ResNet-18 | ResNet-50 | WRN-50 | ResNeXt-50 | Avg. |
|---|---|---|---|---|---|
| Vanilla | 78.45 | 79.30 | 79.35 | 80.86 | 79.49 |
| Fast AdvProp | 78.89 | 79.43 | 79.69 | 79.30 | 79.33 |
| AdvProp | 75.15 | 78.05 | 77.50 | 78.36 | 77.27 |
| Vanilla + MixUp | 79.23 | 79.04 | 80.75 | 81.39 | 80.10 |
| MixProp | **80.86** | **81.84** | **82.17** | **82.37** | **81.81** |
| Fast AdvProp + MixUp | 78.94 | 80.68 | 80.19 | 80.74 | 80.14 |
| AdvProp + MixUp | 77.04 | 79.87 | 80.22 | 80.91 | 79.51 |
| EntProp ($k=0.2, n=1$) | 79.41 | 79.99 | 80.66 | 81.46 | 80.38 |
| EntProp ($k=0.4, n=1$) | 78.99 | 79.69 | 81.14 | 81.46 | 80.32 |
| EntProp ($k=0.6, n=1$) | 78.89 | 80.31 | 81.30 | 81.75 | 80.56 |
| EntProp ($k=0.2, n=5$) | 79.19 | 78.21 | 81.13 | 81.20 | 79.93 |
| EntProp ($k=0.4, n=5$) | 79.81 | 80.59 | 81.51 | 81.64 | 80.89 |
| EntProp ($k=0.6, n=5$) | 78.92 | 80.62 | 80.77 | 81.38 | 80.42 |

| Method | ResNet-18 | ResNet-50 | WRN-50 | ResNeXt-50 | Avg. |
|---|---|---|---|---|---|
| Vanilla | 49.96 | 51.01 | 51.64 | 52.95 | 51.39 |
| Fast AdvProp | 53.31 | 54.23 | 55.25 | 55.31 | 54.52 |
| AdvProp | 56.78 | 58.94 | 59.28 | 59.08 | 58.52 |
| Vanilla + MixUp | 53.84 | 54.03 | 56.68 | 58.29 | 55.71 |
| MixProp | 53.97 | 55.55 | 56.38 | 56.97 | 55.72 |
| Fast AdvProp + MixUp | 54.65 | 57.04 | 57.16 | 57.83 | 56.67 |
| AdvProp + MixUp | 56.34 | **61.06** | **61.46** | **61.53** | **60.10** |
| EntProp ($k=0.2, n=1$) | 55.24 | 56.07 | 57.30 | 58.47 | 56.77 |
| EntProp ($k=0.4, n=1$) | 55.75 | 56.48 | 58.74 | 57.53 | 57.13 |
| EntProp ($k=0.6, n=1$) | 55.86 | 57.41 | 58.95 | 59.28 | 57.87 |
| EntProp ($k=0.2, n=5$) | 54.52 | 57.27 | 58.00 | 58.95 | 57.19 |
| EntProp ($k=0.4, n=5$) | 56.94 | 58.75 | 60.70 | 59.79 | 59.04 |
| EntProp ($k=0.6, n=5$) | **57.16** | 60.50 | 61.02 | 61.35 | 60.01 |

Table 8: Standard accuracy (top table) and robust accuracy (bottom table) for the ResNet family on the CUB-200-2011 dataset.

| Method | ResNet-18 | ResNet-50 | ResNet-101 | ResNet-152 | Avg. |
|---|---|---|---|---|---|
| Vanilla + MixUp | 77.30 | 83.04 | 83.45 | 84.15 | 81.99 |
| MixProp | **79.08** | **83.77** | 83.71 | 84.25 | **82.70** |
| Fast AdvProp + MixUp | 77.88 | 82.69 | **84.05** | **84.34** | 82.24 |
| AdvProp + MixUp | 74.56 | 81.11 | 82.43 | 82.71 | 80.20 |
| EntProp ($k=0.2, n=1$) | 77.64 | 82.92 | 83.85 | 84.11 | 82.13 |
| EntProp ($k=0.2, n=5$) | 77.95 | 83.10 | 83.68 | 84.06 | 82.20 |

| Method | ResNet-18 | ResNet-50 | ResNet-101 | ResNet-152 | Avg. |
|---|---|---|---|---|---|
| Vanilla + MixUp | 51.90 | 58.52 | 62.60 | 63.46 | 59.12 |
| MixProp | 51.32 | 56.80 | 60.36 | 61.90 | 57.59 |
| Fast AdvProp + MixUp | 53.27 | 60.32 | 64.23 | 64.27 | 60.52 |
| AdvProp + MixUp | 54.08 | **62.59** | **65.71** | **66.57** | **62.24** |
| EntProp ($k=0.2, n=1$) | 53.84 | 60.18 | 64.12 | 65.20 | 60.83 |
| EntProp ($k=0.2, n=5$) | **54.58** | 60.71 | 64.09 | 65.44 | 61.21 |

Table 9: Standard accuracy (top table) and robust accuracy (bottom table) for the EfficientNet family on the CUB-200-2011 dataset.

| Method | B0 | B1 | B2 | B3 | B4 | B5 | B6 | B7 | Avg. |
|---|---|---|---|---|---|---|---|---|---|
| Vanilla + MixUp | 82.09 | 83.75 | 83.54 | 84.38 | 84.61 | 84.25 | 84.93 | 84.61 | 84.02 |
| MixProp | **82.86** | **84.16** | **84.61** | **85.16** | **85.73** | **85.40** | **85.53** | **85.71** | **84.90** |
| Fast AdvProp + MixUp | 81.99 | 83.12 | 83.67 | 84.57 | 84.36 | 84.02 | 84.70 | 84.58 | 83.88 |
| AdvProp + MixUp | 78.88 | 79.43 | 79.89 | 80.65 | 81.25 | 80.82 | 81.40 | 81.39 | 80.47 |
| EntProp ($k = 0.2, n = 1$) | 81.97 | 83.01 | 83.44 | 84.04 | 84.42 | 83.99 | 84.40 | 84.41 | 83.71 |
| EntProp ($k = 0.2, n = 5$) | 82.14 | 83.03 | 83.30 | 84.06 | 84.49 | 83.83 | 84.21 | 84.87 | 83.74 |

| Method | B0 | B1 | B2 | B3 | B4 | B5 | B6 | B7 | Avg. |
|---|---|---|---|---|---|---|---|---|---|
| Vanilla + MixUp | 58.68 | 63.36 | 63.38 | 62.86 | 64.47 | 66.25 | 67.34 | 66.76 | 64.14 |
| MixProp | 57.50 | 61.80 | 62.26 | **64.37** | 65.72 | 65.72 | 67.14 | 66.77 | 63.91 |
| Fast AdvProp + MixUp | 59.34 | 63.27 | 63.58 | 63.53 | 63.95 | 66.06 | 67.22 | 67.39 | 64.29 |
| AdvProp + MixUp | 57.68 | 59.59 | 59.79 | 59.93 | 62.05 | 62.60 | 63.81 | 64.03 | 61.19 |
| EntProp ($k = 0.2, n = 1$) | 59.93 | 63.54 | **64.32** | 63.90 | **66.31** | **66.99** | **68.16** | 67.43 | **65.07** |
| EntProp ($k = 0.2, n = 5$) | **60.10** | **63.64** | 63.92 | 63.31 | 65.07 | 66.97 | 67.96 | **68.45** | 64.93 |

Table 10: $H_{score}$ for different data augmentations with ResNet-18 trained by EntProp ($k = 0.2, n = 1$) on the CIFAR-100 dataset.

| Data Augmentation | SA(%) | RA(%) | $H_{score}$ |
|---|---|---|---|
| MixUp | 79.41 | **55.24** | **65.15** |
| CutMix | **81.39** | 50.78 | 62.54 |

