# OpenReview forum: "EntProp: High Entropy Propagation via Auxiliary Batch Normalization Layers"
_ICLR.cc/2024/Conference — Submitted to ICLR 2024_

### Official Review · Reviewer_2wsR · 2023-10-15

**Soundness:** 2 fair
**Presentation:** 3 good
**Contribution:** 2 fair
**Rating:** 5
**Confidence:** 4

**Summary:**

The paper introduces a new approach EntProp for enabling deep neural networks to generalize to out-of-distribution domains. The key contribution of this work is the proposal of selecting clean, high-entropy samples to the auxiliary batch normalization layers (ABNs) to improve the standard accuracy and robust accuracy. This work presents two techniques, data augmentation, and free adversarial training, that increase entropy and bring the sample further away from the in-distribution domain. EntProp effectively disentangles clean and transformed sample domains, leading to higher standard accuracy and robustness with a lower training cost than existing methods. The paper provides detailed experimental results on CIFAR-100 and CUB-200-2011 datasets, showcasing the method's efficiency and ability to prevent overfitting compared to existing techniques.

**Strengths:**

(1) The paper introduces a unique approach, EntProp, challenging the conventional separation of clean and transformed samples as distinct domains. It offers an innovative perspective on domain separation and out-of-distribution robustness.

(2) The paper is well-structured, with clear explanations of the proposed method and the underlying hypotheses, especially in explaining the pseudo-code and details.

(3)The experiments are comprehensive for evaluating the proposed EntProp approach, including ablation studies, sample selection bias analysis, and comparison to existing methods.

(4) This work achieves both high standard accuracy and robustness against out-of-distribution data, which is a critical problem in practical machine learning applications. EntProp's ability to enhance both while avoiding overfitting to adversarial attacks holds practical value.

**Weaknesses:**

(1) The paper lacks a theoretical foundation to support its claims. Providing theoretical insights into why increasing entropy improves disentangled learning and robustness would greatly enhance the paper's clarity.

(2) This work lacks scalability analysis for very large datasets or complex models. Investigating the method's performance on more extensive tasks would demonstrate its practicality in real-world scenarios.

(3) While the paper compares its method with other baselines, it could benefit from a more extensive comparison with state-of-the-art methods[1, 2, 3] in disentangled learning, out-of-distribution robustness, and adversarial training to demonstrate its superiority convincingly.

(4) A more comprehensive analysis of robustness encompassing diverse distribution shifts and attack strategies would strengthen the paper's contributions. Furthermore, the evaluation based on just two datasets (CIFAR-100 and CUB-200-2011) may not be as persuasive compared to other research in the same domain. Expanding the evaluation to include a more diverse set of datasets would enhance the paper's credibility.

(5) The paper lacks an in-depth discussion of scenarios where the proposed method may fail or have limitations. Acknowledging these limitations and providing insights into potential remedies would enhance the paper's credibility.

[1]  Domain Generalization with Mixstyle, ICLR 2021

[2] Out-of-Distribution Detection with Deep Nearest Neighbors, ICML 2022

[3] Adversarial Distributional Training for Robust Deep Learning, NeurIPS 2020

**Questions:**

(1) The paper's evaluation is limited to two datasets, CIFAR-100 and CUB-200-2011. It would be valuable to provide a justification for the selection of these datasets. How well do they represent the diversity of real-world scenarios, and can the findings be extended to other datasets and domains?

(2) While the paper highlights improvements in adversarial robustness, it would be beneficial to include a more extensive analysis covering various types of adversarial attacks to provide a comprehensive assessment of the method's effectiveness.

(3) It would be insightful if the authors could compare their proposed method with other state-of-the-art techniques for adversarial training and robustness enhancement. Such comparisons would offer a more comprehensive understanding of the method's relative performance.

(4) The paper primarily focuses on image classification tasks. Is there potential for the proposed approach to generalize to a broader spectrum of machine learning tasks, such as object detection or natural language processing? Exploring its applicability in diverse domains would be valuable.

---

### Official Review · Reviewer_hjam · 2023-10-30

**Soundness:** 3 good
**Presentation:** 2 fair
**Contribution:** 1 poor
**Rating:** 3
**Confidence:** 3

**Summary:**

This paper falls in the field of robustness. More specifically, it argues that some samples after applying some transformation will still be "in-distribution'' based on the sample's entropy. The authors introduce two techniques to increase the entropy and push the sample away from the "in-distribution" domain. Experiments are conducted to show that the proposed techniques can help.

**Strengths:**

1. This paper has a good motivation to improve the robustness of trained models.
2. The paper is generally well-written.

**Weaknesses:**

1. Correct me if I am wrong, but in my opinion the researching question seems to be constantly true. It is almost certainly that some transformations result in in-distribution samples, for example translation and rotation. It would make more sense if there are more constraints on types of transformation.
2. I believe the contribution bullet no. 3 shall be merged with bullet no. 2 since they are all describing similar outcomes, i.e., EntProp improves the standard and robustness accuracy. In such a case I think the contribution of this manuscript is not sufficient.
3. I would suggest a better structure of Related Works. For example, it would be better to separate architectures (like multiple BNs) with adversarial training methods (augmentation, adv attacks etc.).
4. I have concern about the novelty of this manuscript: it seems that all the techniques are just taken from existing works. MixUp is well-known and the Fast Adversarial Training is from Shafahi et al. (2019).

**Questions:**

1. Looking at Table 3, it seems that the performance generally goes up when $k$ increases. What is the performance when $k$ equals $0.8$ or $1$? Can they achieve better performance? If yes, then the subsampling process could be not effective at all.
2. In Table 2, what is the performance without sample selection (i.e., the simpest baseline? )
3. In Table 4, why not conduct experiments with k=0.6, which has the overall highest performance in Table 3? Is there any specific reason for this choice?
4. How to pick $k$ in principle?

---

### Official Review · Reviewer_CJYg · 2023-10-31

**Soundness:** 2 fair
**Presentation:** 2 fair
**Contribution:** 2 fair
**Rating:** 5
**Confidence:** 3

**Summary:**

This article presents a learning framework that aims at improving out-of-distribution performance. Based on AdvProp and MixProp, it proposes to use Main or Auxiliary batch normalization layers (MBNs/ABNs) that learn on separated distributions of data. Based on data augmented on MixUp, the method considers high entropy samples that are also augmented with adversarial examples to be out-of-distribution and trains them on the ABNs.

**Strengths:**

The article is well-written and easy to follow.

The experiments and ablation studies allow a good understanding of the effects of the method and to compare it to the others (with some caveats). In particular, Table 3 showcases well that for a given compute budget, EntProp improves the robustness of the network in most cases.

Entropy seems like a natural metric to distinguish the samples and is well correlated with both augmentations.

**Weaknesses:**

**Results** Table 7 shows that the standard accuracy of the network decreases significantly with EntProp, thus making less obvious the choice of EntProp over the other techniques, in particular MixProp. There is only two different values of $k,n$ for Table 4 compared to Table 3, making it difficult to see their effect. In Table 3, it seems that increasing $k$ increases the accuracy. However, the Table stops at $k=0.6$, while it seems important to increase the value up to $1$ to see where the improvement stops.

**Mixup** Mixup is defined as "the most typical data augmentation" which is surprising. Seeing as all experiments on done on images, there are way more common data augmentation techniques, such as random crop and random flip. Note that these augmentations are done in MixProp. Are those more common data augmentations used in the experiments here?

The citation for Mixup is also very surprising, seeing as it was introduced by Zhang et al. 2018 and not Pang et al. 2019. Furthermore, in neither Zhang et al. 2018 or Pang et al. 2019 is the mixup loss defined like in this article. The labels in mixup are also mixed and a single loss is applied, which is different from the definition used in this paper where two losses on both clean labels are applied and then mixed. This seems to however be also the case in MixProp, which also does not explain where this new MixUp loss comes from (which seems to appear in several other articles that are not cited in either MixProp or EntProp).

**Novelty** This article presents very limited novelty. This is a combination of AdvProp and MixProp, but using the mixed-up examples also in the MBN, and only a percentage of examples in the ABN.

**Metric** The choice of the Harmonic Mean between SA and RA seems a bit arbitrary since it does not seem to be used in other articles.

**Adversarial robustness** MixUp decreases strongly adversarial robustness, however, this is noted by the authors as being different from their wanted metric of out-of-distribution robustness.

**Entropy** Entropy is never defined precisely in the article, how is it measured in practice?

**Other metrics** Why not choose the Confidence metric rather than Entropy? It gives consistently better results than Entropy (except for ResNeXt-50). I am also surprised that the adversarial robustness for the Random metric is given in Table 5, but not the accuracy, which seems significant to see the improvement due to the metrics.


**Hyperparameter** How are k and n chosen in practice? By cross-validation? This is never very clear in the article.

**Questions:**

* Why do feeding random samples (Fig 3) to the ABN increase the accuracy of the network? Since the samples are random, it would seem to me that the ABN would learn the same statistics from the MBN, resulting in similar accuracy. Or I may be mistaken and adversarial examples are being used in what I thought was an ablation experiment because $n$ is not specified.

* The variance of the metrics seems to increase a lot with Mixup in Table 2. This is not elaborated upon as variance is not given in further Tables.

* As $k$ increases, the number of examples in the loss increases, artificially increasing the batch size per iteration. Why is there no normalization taking into account this increase of batch size? (Or other changes)

* Why do the authors think that "AdvProp consistently shows significantly lower results than vanilla training" on the CUB-200-2011 dataset, but consistently the best or second best results on CIFAR-100 ? This seems like a surprising decrease.

* Figure 1: In this case, are transformed samples equal to mixup samples for MixUp and adversarial examples for AdvProp? This is not clear in the text.

* Entropy seems to decrease in training. Why do the authors think this happens?

---

### Official Review · Reviewer_A6wK · 2023-11-01

**Soundness:** 2 fair
**Presentation:** 2 fair
**Contribution:** 2 fair
**Rating:** 3
**Confidence:** 5

**Summary:**

The authors propose a novel approach called High Entropy Propagation (EntProp) that builds on disentangled learning with mixture distribution via Auxiliary Batch Normalization layers (ABNs). The key insight is that some transformed samples are not completely different from clean samples. EntProp identifies clean high-entropy samples and applies transformations to push them further into the out-of-distribution domain.

**Strengths:**

1. identifying and transforming high-entropy samples to generate out-of-distribution samples is an interesting approach, and also an intuitively efficient method.

2. The idea that EntProp can prevent overfitting on small datasets is quite interesting, this is often an under-studied topic.

3. The discussion on computational efficiency is helpful for one to better appreciate the method.

**Weaknesses:**

1. the emprical scope of this paper is unfortunately too limited. At this moment, it is probably too far for being considered at this venue. For example, the authors' writing suggests that the authors are contributing to the line of research in robustness (maybe particular with data augmentation), then there are tons of methods that are along the robustness research, and many of them are using data augmentations. There are many other OOD benchmarks, like ImageNet variants, many corruptions datasets, like -C datasets, and many other adversarial attacks. The empirical coverage of this paper is too limited that it unfortunately does not touch any of the above directions sufficiently.

2. The main idea of this paper is quite similar to [1], in the sense of first identifying the samples with high entropy (high loss) and then perform worst-case data augmentation to those samples (adversarial attack or dropout-style augmentation). It could be interesting if the authors offer more in-depth discussion of the relationship.

[1]. The two dimensions of worst-case training and their integrated effect for out-of-domain generalization

**Questions:**

1. It is unclear to me why the authors choose to work on these two datasets instead of more standard ones.

2. When the authors make claims like this "In this paper we focus on improving both standard accuracy and robustness against out-ofdistribution domains." one will typically expect much stronger empirical results. It seems the authors might want to significantly re-write the scope of this paper. Maybe it's only about methods that are explicitly dealing with augmentation, OOD, and BN.

---

### Meta-Review · Area_Chair_72Ev · 2023-12-11

**Metareview:**

This paper presents a novel approach named EntProp, aimed at enhancing out-of-distribution robustness. The method utilizes Auxiliary Batch Normalization layers for managing high-entropy samples, potentially offering a balance between standard accuracy and robustness. Despite some innovative elements, reviewers have raised several major concerns about the paper's empirical scope, novelty, methodological clarity, comparative analysis, and citations. Unfortunately, the authors did not provide a rebuttal addressing these concerns.

The AC encourages the authors to carefully tackle these concerns and make a stronger submission next time.

**Justification For Why Not Higher Score:**

The authors of this paper did not provide a rebuttal to address the concerns raised by reviewers.

**Justification For Why Not Lower Score:**

N/A

---

### Decision · Program_Chairs · 2024-01-16

Reject